# Situating Stigma: An Ethnographic Exploration of How Stigma Arises in Interactions at Different Stages of People’s Drug Use Journeys

**DOI:** 10.3390/ijerph20196894

**Published:** 2023-10-07

**Authors:** Fiona Catherine Long, Kirsty Stuart Jepsen

**Affiliations:** School of Social Sciences, Cardiff University, Cardiff CF10 3AT, UK; stuartkf@cardiff.ac.uk

**Keywords:** ethnography, situated stigma, identity, drug use

## Abstract

The association between stigma and drug use has been widely researched. However, to fully understand the nuances of stigma, as experienced by people who use drugs (PWUD), it is argued that we must look at the situations within which stigma is encountered. To obtain an ‘up close’ look at situated stigma, data are drawn from two ethnographic studies—one set in a homeless hostel in the South of England and the other at a substance use service in South Wales. This article explores how PWUD experience and negotiate stigma at different stages of their drug use. We identify three notable themes across these settings. Firstly, ‘othering’ occurs in two distinct ways, by othering the past self or distancing from other PWUDs. Secondly, ‘drug exceptionalism’ is used to justify an individual’s drug use and express frustration at the associations between legality, social harm, and stigma. Finally, in ‘negotiating identities’, individuals present alternate identity roles to either demonstrate clashes in identity or to promote a conventionally positive sense of self. This article contributes to the existing literature on stigma, firstly, by comparing the stigma management strategies of those in active drug use and recovery and, secondly, by using this to highlight the importance of ethnography and situated stigma within this field.

## 1. Introduction

People who use drugs (PWUD) are familiar with the stigma of being seen as a ‘junkie’, or an ‘addict’ [1,2,3]. According to Livingston et al. [4], stigma manifests itself in three principal ways: social stigma embodies stereotypes and prejudices endorsed by others; self-stigma represents internalised feelings of shame and low self-efficacy; and structural stigma refers to prejudice and discrimination in policies, laws, and institutions. PWUD commonly experience social stigma through their interactions with others, which when internalised, can become self-stigma, leading to potentially detrimental effects for the individual [4,5,6,7].

As drug use has the potential to be ‘deeply discrediting’, by marking individuals as less desirable, drug users may employ a range of stigma management strategies to alleviate the effects of this stigma [8]. This need is rooted in the presentational demands of the self in social situations, as individuals’ social selves only exist through social interaction [9]. Stigma management strategies are multiple and varied. For instance, Snow and Anderson [1,10] identify three forms of identity work amongst those experiencing homelessness, of which ‘distancing’ was one. They note that individuals may attempt to distance themselves from roles, associations, or institutions, which ‘imply social identities inconsistent with their actual or desired self-conceptions’ [10] (p. 215). Individuals may therefore actively stigmatise others in order to deflect attention from their own stigma, creating distance between the ‘drug-using self’ and ‘junkie other’ [11]. This is possible because ‘the normal’ and ‘the stigmatised’ are not people but perspectives, meaning that within interaction, ‘he who is stigmatized in one regard nicely exhibits all the normal prejudices held toward those who are stigmatized in another regard’ [8] (p. 138). This means that drug users may experience stigma, whilst simultaneously stigmatising others. For example, small cliques may define their own drug use as normal, when compared to the perceived abnormal use of others, resulting in the creation of drug use hierarchies within interactions [3,12,13].

Distancing, as a stigma management strategy, does not only occur in relation to others, rather it is also possible to distance from one’s former ‘drug-using’ self, something which is particularly evident amongst those engaging with drug use services [14,15]. These individuals often embrace the disease model of drug use and the ensuing necessity for treatment, which suggests that identity construction reflects specific ideals that are rooted in policy, such as the need for self-regulation [15,16].

In employing these strategies, individuals may make the case for ‘drug exceptionalism’, by suggesting that some illegal drugs are less socially and physically harmful than others and therefore should either be considered less morally repugnant or even legalised [17,18,19]. These arguments also extend to legal drugs and alcohol, given the well-documented mismatch between legality and potential harms [20]. However, singling out certain drugs as acceptable via negative comparison perpetuates the stigma associated with drug use more broadly [3].

A final concern for PWUD relates to their negotiation of potentially conflicting roles and identities. In some cases, PWUD actively invoke prosocial roles—roles which are widely considered positive—in the face of stigmatising drug user identities, such as ‘junkie’ or ‘addict’ [2]. Prosocial roles may relate to positive past identities, such as past employment or achievements [21], or future goals and the roles they entail, including employment, money, possessions, and romantic partners [1], in other words, the ingredients of a ‘normal’ life [16]. However, this strategy is not without its problems. Firstly, individuals contained within institutions may struggle to foreground prosocial identities, as they are constrained by their immediate setting, given that identity roles must be relevant ‘for this setting and for this activity’ [22] (p. 342). Secondly, prosocial roles may conflict with drug-using roles in multiple and complex ways, as is evident in the context of motherhood, as mothers who have used drugs attempt to navigate and even restore their ‘spoiled’ identities [14,23,24].

Whilst associations between stigma and drug use are well documented in the academic literature [25], insufficient attention has been paid to actual observed manifestations of stigma, in terms of how it presents itself and is negotiated by PWUD. This is significant, as whilst drug use has the potential to be ‘deeply discrediting’, it may equally ‘confirm the usualness’ of an individual, depending upon context and, more specifically, upon the interaction within which it arises [8].

As stigma is an interactional phenomenon, the present study seeks to understand how stigma arises and is managed by PWUD within the context of situated interactions. In particular, we seek to comprehend similarities and differences in stigma management strategies from individuals engaged in active addiction to those who framed themselves to be in recovery. In doing so, we aim to not only shed light on stigma management strategies themselves but also the pervasiveness of stigma throughout individuals’ drug use journeys. This article contributes to the existing literature on stigma in two ways: firstly, by comparing the stigma management strategies of those in active drug use and recovery and, secondly, by using this to highlight the importance of ethnography and situated stigma.

## 2. Materials and Methods

The findings presented in this paper are drawn from two ethnographic research projects that involved PWUD. One of the key benefits of taking an ethnographic approach is it is often foregrounded by a period of observation, which takes place in a ‘natural’ setting, thereby enabling researchers to immerse themselves in the lives of their informants, by listening to what they say, watching what they do, and situating this within a specific context [26,27]. Moreover, ethnography facilitates a holistic understanding of topics and populations under study, creating space for the messiness and subtleties of social life, therefore doing justice to the stories of participants [28]. Ethnographic approaches typically employ multiple data collection methods beyond observations, in order to better understand the lives of informants.

Whilst ethnography has faced some criticism, particularly related to its perceived subjectivity, these tend to centre on positivistic measures of validity, reliability, and generalisability, which encompass key epistemological and ontological differences [28]. As such, the research involved in these studies adheres to alternate qualitative measures of credibility, which rely practically on thick description, audit trails, data-led findings, and plausibility, verified by member checking and the triangulation of methods [29]. Both research projects adhered to these criteria and employed additional qualitative methods to verify the findings.

This approach enabled the researchers to observe the subtle and nuanced ways in which stigma presented itself and was consequently negotiated by PWUD *in situ*. The two datasets engaged individuals at different stages in their drug use journey: the first with those in active addiction and the second with those in recovery. By comparing these datasets, we were able to trace similarities and differences in experiences of stigma, as they occurred at different stages in the drug use journey.

The first ethnographic study took place at a large male-only homeless hostel in the South of England, which has been given the pseudonym Holbrook House (HH). Having once worked at the hostel, the researcher sought to put distance between herself and the setting by leaving her role 9 months prior to data collection, whilst reflexively accounting for any lingering familiarity with the place, people, and processes within it. The short-term nature of hostel accommodation—with a suggested stay of 6 months—meant that the resident turnover was high. This hostel was categorised as ‘high support’, which reflected the multiple and complex needs of its residents, one of which related to drug and/or alcohol use. Whilst the researcher had not set out to study drug use, the prevalence of drug and alcohol use amongst residents meant that consumption formed a routine aspect of daily life at the Holbrook House and therefore became central to the study.

Data collection occurred between January 2020 and November 2021, with several interruptions due to the onset of the COVID-19 pandemic. The primary data collection method used was in-person participant observation (218 h), although a number of semi-structured interviews were conducted with current residents and members of staff (*n* = 17), as well as creative timeline interviews with former hostel residents (*n* = 3). Consent was obtained verbally for each method, following conversations regarding what exactly that method would entail, although additional written instructions were provided ahead of timeline interviews for clarity. The researcher developed rapports in the field through frequent mundane chit-chat and by simply hanging around. Whilst nobody explicitly refused to consent, some residents were clearly unwilling to participate during the participant observation, as was evident from their one-word answers and tendency to avoid the researcher. Timeline interviews were undertaken by a snowball sample of former Holbrook House residents. Participants were simply asked to ‘create a timeline of your experiences of homelessness,’ thereby giving them the freedom to map their own journeys and organise their own experiences. The timelines were used as an elicitation device in interviews with the researcher, who then transcribed and analysed the interview data.

The second study was set at a drug use service based in South Wales, with the pseudonym Diwrnod Newydd (DN). DN presented itself as a peer-support recovery community, which aimed to empower its members to achieve and maintain recovery from problematic drug and alcohol use. The definition of ‘recovery’ used in this article reflects that of the service and members of DN. DN presented recovery as achieving a life free from problematic use, and although the service advocated for abstinence, they recognised that this is not how everyone defined their recovery; therefore, abstinence was not a prerequisite to becoming a member. Participants were members of this community and ranged from those at the start of their ‘recovery’ to those who had been a part of this community for many years.

Access to DN was gained via a gatekeeper, and overt participant observation took place between June 2021 and May 2022. The researcher brings knowledge of the sector in their additional role as a mental health nurse; however, within this study, the researcher took the position of an ‘outsider’ due to having no prior relationship with the organisation and having no experience of being in ‘recovery’. During this time, the service offered groups online due to the COVID-19 pandemic; therefore, participant observation not only took place offline but also online via the platform Zoom (263 h). The meetings via Zoom had a pre-set weekly schedule, with each group pertaining to a pre-set topic or type of support (e.g., peer support or social group), with an aim to mimic the group support that would have previously occurred face-to-face. At the beginning of every meeting, each attendee was invited to ‘check in’, which provided a natural opportunity for the researcher to disclose their position to the group to ensure overt observation occurred. Participant observation was accompanied by ‘photo elicitation’ conversations with several male members of the service (*n* = 7) who were recruited from within DN by way of convenience and snowball sampling. Those who consented to take part in a conversation about their recovery were asked to bring photographs along with them that they felt were relevant to their personal and social identities and helped to explain their recovery journey. These photographs were used as a tool to help communicate their experiences and as a prompt for additional topics of conversation. Conversations were audio recorded, transcribed verbatim, and analysed by the researcher, and all participants were asked for additional consent for their photographs being used by the researcher. Informed consent was obtained verbally for each method, following disclosure of the researcher’s position and verbal information about the study. Due to the length of time that observational data was collected, consent was treated as an ongoing process, rather than a one-off event, which lent itself to asking for verbal consent throughout the data collection period as part of an ongoing conversation. No one refused to consent to be part of the study; however, only men consented to take part in additional photo elicitation conversations.

The datasets were analysed separately, as part of two distinct doctoral research projects, each comprising a series of fieldnotes and interview transcripts. Both projects employed Charmaz’s [30] constructivist grounded theory, which ensured an inductive approach to data analysis. This involved an iterative coding process, the creation of analytic memos, the identification of recurrent themes, theoretical sampling, and ultimately the construction of a theory. In comparing the two datasets, the researchers used the themes that were common across both datasets as a starting point. The data that made up these themes were then compared and contrasted, in the search for overlaps, distinctions, and nuances between the two. After several rounds of comparison, shared themes were refined, and three key areas were developed for joint analysis: namely, distancing, drug exceptionalism, and negotiating identities. All individuals and services have been anonymised and given pseudonyms.

## 3. Results

### 3.1. Distancing

Residents at HH find that their identity is restricted by the hostel itself, and as such, they employ techniques to discursively distance themselves from their peers, as a means of salvaging the self. Those who do not use any drugs take a similar line, ‘I shouldn’t be here too long, I’m not a druggy’ (Fieldnotes, 4 November 2020), implying an overlap between residency at HH and drug use. Those who do use drugs often mitigate their own use and choice of drug, by condemning those of others. This is apparent in the way that those who engage in intravenous (IV) use (such as crack and heroin) talk about those who use spice (spice is a synthetic cannabinoid, though it is often much stronger and more unpredictable than marijuana) and vice versa.


*There’s shouting in the hallway, “I need my fucking money. I’ll come to your house and take your TV”. Noah says, “that’s spice for you, those who don’t fit out get aggressive”. Noah is an IV user, though is “really anti-spice”, he’s seen people fitting, convulsing and killing themselves over it. (Fieldnotes, 24 January 2020)*



*Isam, a frequent spice user, warns me about “heroin men”, “they’re liars, don’t trust them”. A new resident joins in, “are you talking about crack?” Isam said yes, “I know crackheads who’d put a blade into someone for one pipe”. (Fieldnotes, 15 September 2021)*


Through the process of distancing, residents are effectively othering and stigmatising other groups of individuals within the hostel, whilst reaffirming that they are not at the bottom of the drug use hierarchy. Furthermore, those using the same drugs similarly distance themselves from others, based on their using practices. Take those who engage in IV use for example:


*Noah criticises rough sleepers for injecting into their feet, “blatantly in front of everyone in town”. (Fieldnotes, 24 January 2020)*


*Shaun shows me his veins and points out that they are all still there, aside from a little bruising. He rotates when he injects and doesn’t inject into his legs because this can cause ulcers. He says people damage their veins by being “lazy and using old blunt needles from the drawer”. (Fieldnotes, 20 February* 2020)

Residents at HH effectively create drug hierarchies through situated interactions. These hierarchies are situated insofar as they are not fixed but malleable, adaptable to individuals and situations, with the purpose of demonstrating that their drug of choice or drug use practice is never the worst.

For those who live at the HH but no longer use drugs, distancing is still evident, although the tone is much more sympathetic:


*Leo says, “I look at most people in here and think I’m glad I’m not there… I’ve been there and I’m glad I’m not there anymore”, “I look back and think it was too much effort”. (Fieldnotes, 20 July 2021)*


Moreover, once individuals move out of the hostel and into independent accommodation, they reflect upon their past experiences and distance themselves from their former drug-using self. Paul describes, ‘bringing back the Paul pre-drugs and pre-prison’:


*I had reached a point where my life needed to change in order to save my life, and rebuild what I had so recklessly destroyed through drugs and criminality. And now four years substance free, I’m in a good place going forward from here, and always remembering I am a new one seemingly irrelevant decision from spoiling all that. (Interview with Paul)*


Within the hostel, residents do not stigmatise or distance on the basis of the ‘homeless’ label, as they are all in a similar position. However, beyond its walls, it may become difficult to distinguish whether stigma is attributable to homelessness or drug use.

At DN, members did not distance themselves from drug use via comparisons with others who use drugs. Similar to those who had moved on from HH, members of DN focused on their own past experiences and spoke about how their views and actions had changed. A ‘new me’ (Luke, photo-elicitation interview) was spoken about as a way to separate past actions from a current positive self, with a ‘then versus now’ narrative reinforced by peers; ‘you are not that person anymore’ (Fieldnotes, 16 August 2021). When talking about negative past behaviours, members conveyed regret which continued to affect them although currently abstinent:


*Linda shared that both her children were born dependent on heroin; ‘The guilt associated with this makes me overcompensate in other ways. Guilt is the hardest thing to deal with” (Fieldnotes, 28 July 2021)*


Although feelings of regret and guilt must not be universally conflated with self-stigma, there are times when societal stigma is internalised and reproduced. Dai spoke about the self-stigma felt when he went to prison due to his drug use, a stigma that he then reproduced and justified as fair within society:


*“So, I had my prison haircut ready. I’m laughing about it now, but it wasn’t. Seriously bad dark days they were. So. And the stigma, you create yourself and yeah. DBS (Disclosure and Barring Service (DBS) checks contain details of past convictions and unspent cautions, and may be run for a number of reasons, including employment) is, yeah, it’s grim for everything, the bank, Law Society, nursing. No. Primary teaching. I can’t teach primary kids if they’re under 4, because I’ve had a custodial sentence. And you know people, people say, oh no that’s not fair, it is! If, if, if it was my kids and they were four… No, I wouldn’t want somebody who’d been to prison. Not because I’m judging them. It’s just the fact that, well, no, that’s it, end of really. Might be the nicest guy in the world, but nope sorry”. (Dai, Photo-elicitation conversation)*


Dai’s reasoning as to why a custodial sentence should restrict occupational opportunities demonstrates his belief that what he did in the past was ‘wrong’ and should have continued consequences, despite the changes he had made in his life. Despite distancing from their past selves, members of DN voice that the ongoing consequences of their actions are fair and just within society. Therefore, the ability to distance oneself from prior drug use is not only a means of managing stigma but a process that contributes towards identity reparation and the construction of a prosocial identity.

There are a few occasions in which members of DN discursively distance themselves from other PWUD; however, they occur whilst reflecting on their own beliefs, which were barriers to help-seeking:


*“I was always so stubborn-minded, ‘I’m not one of these people’, you know. Alcoholics are people who are on a park bench. They smell, they do this they, they, they, you know. I had this perception of what an alcoholic was, and how wrong was I? […] I was just like, that’s not me. I work. I’ve got a mortgage. I’ve got this, I’ve got that, that’s not me’. (Luke, Photo elicitation conversation)*


Upon reflection, members at DN appear to recognise the dangers of ‘othering’ those who use drugs and comprehend how societal stigma delayed their own help-seeking. Becoming a part of this ‘recovery’ community means that stereotypes are broken down, and members actively take care to reduce any judgement towards others.

Members of HH and DN both distance themselves from drug use in order to maintain a positive sense of self, albeit in two distinctive ways. When engaging in active drug use, residents mitigate their own use by othering their peers’ use. However, further along on the drug use journey, focus generally shifts from others’ use to individuals’ own past experiences, as the harms of stigma are reflected upon. Those with prior experience of drug use discursively distance themselves from their past self; however, feelings of shame were still evident, challenging their ability to completely manage felt stigma.

### 3.2. Drug Exceptionalism

The aforementioned hierarchies are not limited to illegal drugs, rather residents at HH seek to expand the hierarchy to include more socially acceptable addictions:


*Shaun says, “you’re always gonna chase something, chase money, women, cars”, Vince adds, “everybody has habits, if you don’t go for women, you’re a drunk, if you’re not a drunk you gamble, and that’s the worst”. (Fieldnotes, 3 March 2020)*


It is common for residents to stigmatise legal substances—namely alcohol and prescription medications—by drawing them into this hierarchy. Those who use drugs are critical of those who drink alcohol, and often position them as being a drain on the NHS or society at large.


*Anthony is on the phone. An ambulance pulls up shortly afterwards and takes him to a private room around the corner. After they leave, Jake announces that he is “wasting their time” and that there are “people out there dying”, all he needs is another drink or a detox. (Fieldnotes, 12 March 2020)*


Furthermore, prescription medications, such as methadone and pregabalin (pregabs), are sometimes positioned as being similar to or worse than the drugs taken by many of HH’s members. Some joke about the fact that methadone is used to treat opiate addiction, though both are class A drugs and equally addictive, often leading to simultaneous heroin and methadone addictions. Others position these prescription drugs as dangerous and consequently place them at the bottom of the drug hierarchy.


*Isam maintains that doctors are killing people by prescribing pregabalin. When you take spice on its own it’s okay, when it’s taken alongside pregabs, “your body closes down”. Another man adds that the recent string of deaths in the park were due to people taking pregabs and smoking spice, “it’ll put you to sleep, won’t fucking wake up mate”. (Fieldnotes, 11 March 2020)*


By bringing ‘legitimate’ substances into the hierarchy, the hostel’s residents question the arbitrary lines drawn between legal and illegal substances and suggest that this division is not necessarily rooted in the actual social harms caused.

Members of DN routinely express the idea that the specific substance is not the problem but that their problematic use was the ‘symptom’ of something else; ‘It’s only when addiction can be seen as a symptom that you can begin to recover’ (Fieldnotes 21 August 2021). Therefore, a hierarchy of drugs is not used to ‘other’ but with respect to the perceived personal and social harms which they cause. Similarly to residents at HH, members at DN often criticise alcohol for its legality and social acceptability compared to that of other ‘less harmful’ illegal drugs, and how its perceived safety and less stigmatising use came from the harm that the prohibition of drugs brings:


*Antonio shows a lot of anger towards the fact that illicit drugs are seen as a problem, “but you can go buy a litre of vodka from the shop and no one would blink an eyelid” (Fieldnotes, 17 September 2021)*


Although not commonly voiced, during a group conversation a preference for legalisation is discussed. The group members talk about each person’s drug use history in a slightly glorified manner, and voice the opinion that amphetamines became more ‘dangerous’ when they had to be obtained on the illicit market, following the prohibition of GP’s prescribed ‘purple hearts’ (Drinamyl tablets or ‘Purple hearts’, were a drug containing amphetamine and barbiturate, which were prescribed by GPs in the early 1960s for weight loss and/or to help treat mild depression and anxiety). Interestingly, this point is backed up by the argument for legalisation on the basis of increased taxation, as a ‘better option for the government’, rather than a harm reduction approach. This is possibly due to the perception that money and regulation are more compelling and socially important arguments within wider society than compassion and harm reduction for PWUD.

Residents at HH feel that legalisation will have a more direct impact on the daily lives of hostel users in terms of their own access to support and a reduction in the social harms associated with drugs:


*Abbas suggests that the government “legalise drugs.” Max agrees, “that would stop all the madness here, shoplifting, prostitution, and put an end to the small dealers”. (Fieldnotes, 10 March 2020)*


In the UK, drug categorisation is not based on the personal and social harms they cause, though their legality consequently creates differences in how PWUD are treated within the legal system, as well as within society. Members of HH and DN both invoke a social harms’ perspective to argue that illegal drugs are less socially harmful than legal substances, such as alcohol. Further, discussions in these two contexts highlight the potential financial and criminal justice benefits of legalising drugs.

### 3.3. Negotiating Identities

A third way in which the residents of HH resist the stigma of being a ‘junkie’ or an ‘addict’—rather than displacing it onto others—is by negotiating their own identities. There are numerous instances in which residents actively portray themselves in prosocial roles, either past or present, by referring to their work, family, or skill set, amongst other things.


*Lincoln drinks and uses a miscellany of drugs. He talks fondly about his kids, and despite seeing them rarely, claims to be a good dad for putting money away to buy them presents at Christmas. (Fieldnotes, 26 July 2021)*



*This week the cooking group are making a fry-up. Samir tells the others that he used to cook breakfasts with his mum at a local café, “I do the best breakfasts, used to work 6 till 6, cooking the same meal all day.” (Fieldnotes, 11 February 2020)*


Furthermore, many residents hold largely conventional goals, which are remarkably similar to those held by the rest of society.


*Shaun places his head in his hands, “I want a normal life more than anything. A flat, a job, a car, a missus, a little girl, a holiday once a year, that’s what I want, but I can’t stop using.” (Fieldnotes, 3 March 2020)*


However, these attempts at negotiating identity are constrained. In the former example, role relevancy is limited by setting and activity; therefore, it is difficult to claim the role of ‘good dad’ whilst living at a homeless hostel. In the latter, there is a clear gap between aspirations towards and attainability of these goals.

As explored above in the theme ‘distancing’, members at DN exhibit shame in their talk when speaking about past actions that conflict with other identities, such as parenthood and/or careers.

*“[My son] probably would have wanted to do something else if I asked him. But yeah, just like took him to the pub with me. And er, yeah and then we ended up staying up in the pub like, two in the afternoon till 10 o’clock at night. It’s something I’m so ashamed about.” (Luke, photo-elicitation convers*ation)

Luke speaks emotionally about past actions surrounding his son, stating that he was always there materialistically for his son but not emotionally, such as taking him to the pub for the day and ‘always finding an excuse to drink’. He projects the notion of moral decision making in his past judgments that contribute towards his feeling of having been a ‘bad father’. When this conflict in identity is highlighted, it is used to reinforce the social importance of their recovery:


*“That was the first [concert] I’d been to, but, since it was with my son, [drinking] never came into the equation and did I miss it? Sweepingly. Very briefly, maybe. But then I just put the tape fast forward Okay, if I did, and it would smell, and what, what exactly am I getting out of it? Nothing. All I’m doing, all I would be doing is hiding the fact that I’ve had one. What’s the point of that? So, no. Then it was a no-brainer. […] So, if I, if I was to, I’d be jeopardising everything, not just, okay so I’m gonna, I’m gonna have a glass of wine because I’m at a concert and then I’m not going to tell anybody, even though they’ll know. My son will know. So, to say that I’ll be back to square one, is an understatement and how much would I hate myself? Even though people say, ‘Oh no don’t do that, it’s okay’, no it’s not okay. It’s not Okay. And I probably die.” (Dai, photo-elicitation conversation)*


The problem with emphasising prosocial identities as a reason for not using drugs centres on the consequences of a potential relapse. A relapse would create more shame around conflicting parental and drug-using identities and reinforce problematic notions of ‘good versus bad’ father. As discussed previously, the identity negotiations of mothers as a means of reducing stigma and distancing from prior drug use has been highlighted. However, the ways in which men negotiate fatherhood is less researched, though it is shown to be just as important to the members of DN who are fathers.

Whilst engaging in active drug use, the residents at HH attempt to negotiate their identities by invoking prosocial roles and normative goals, albeit not unproblematically. Once in recovery, members of DN highlight prosocial identities to signify the importance of their recovery; however, they continue to feel tension between conflicting drug and prosocial identities, which are difficult to reconcile. Both settings highlight the role of fatherhood in negotiating identities, although in different ways.

## 4. Discussion

This research illustrates the pervasiveness of stigma, as experienced by PWUD, throughout different stages of the drug use journey. In attempting to salvage the self, PWUD demonstrate both an awareness of the stigma associated with drug use and an ability to negotiate that stigma [10]. By taking an ethnographic approach to both contexts, it has been possible to observe the subtleties of situated stigma, in terms of both its manifestations and consequent stigma management strategies [26,27,28].

Similar strategies are employed amongst members of HH and DN, albeit in slightly different ways and to different degrees. At HH, occurrences of stigma are more prevalent, amplified by the ubiquity of drug and alcohol use within the hostel context and the institutional and associational identities that this implies [10,22]. At DN, occurrences of stigma talk are less common, rather stigma presents itself in much more subtle and nuanced ways amongst those in recovery. As such, HH residents’ stigma management strategies are much more overt and direct than members of DN, which arguably reflects the stages of individuals’ drug use journeys.

Firstly, whilst distancing practices are evident amongst members of HH and DN, these practices manifest themselves in different ways. At HH, residents other and stigmatise the drug choices or drug use practices of their peers, which enables them to deflect from their own use, alleviate the stigma they experience, and ensure that they are never at the bottom of the drug use hierarchy [3,11]. Conversely, in creating narratives of a ‘new me’, members of DN are able to distance from, and stigmatise, their past drug-using selves [15]. This enables members to simultaneously manage stigma and construct a new more prosocial identity [2,15].

Members in both settings challenge seemingly arbitrary distinctions between the legal and illegal substances. In doing so, they adopt a social harms perspective, by ranking drugs and alcohol not according to laws but the personal and social harms, which they can cause [20]. In both contexts, this is evident when PWUD talk about alcohol. Whilst alcohol is largely seen as socially acceptable, it is also capable of causing great levels of harm to the person drinking it and to society at large. Whilst Nutt et al. [20] similarly found alcohol to be more harmful than any illegal drug, the assessments of these individuals are not rooted in research but personal experience, as they comment on the ease of buying a litre of vodka or the amount of NHS time wasted by those who drink alcohol. However, such arguments often invoke the discourses of ‘drug exceptionalism’, in which individuals make the case for one drug at the expense of another [17,19]. This can result in other drugs or PWUD being stigmatised by negative comparison, which has the adverse effect of perpetuating stigma associated with drug use more generally [3,18,19]. Furthermore, whilst members of both settings advocate in favour of legalisation, those at DN make economic arguments and position it as the best option for the government, whilst those at HH feel that it would benefit their day-to-day lives, by ‘stopping all the madness’ within the hostel.

The two stigma management strategies considered so far involve shifting and relocating stigma onto other drugs and drug use practices. The final strategy involves individuals looking at themselves and their own multiple identities, as a means of negotiating the stigma attached to the role of a drug user. At HH, residents seek to negotiate and neutralise their own identities in multiple ways, for example, by emphasising prosocial roles and conventional goals [18], for instance, being a ‘good dad’ or wanting a flat, job, car, and a missus, regardless of whether they are past, present, or future [1,10,21]. However, residents find that they are necessarily constricted by the institutional setting within which identity negotiation takes place [22]. Moreover, some individuals may have simply been regurgitating the goals of a ‘normal life’, which were implicit in many programmes [16]. Those at DN distance themselves from the stigma of drug use by separating their former and current ‘selves’, and highlighting how drug use conflicts with who they are now. The context of their current life may protect members from discourses of stigma, as they surround themselves with others who are encouraging and collectively work towards understanding and accepting themselves. However, this distinction between their past and present may cause internal conflict and bring about feelings of shame, most notably in distinguishing the ‘good versus bad father’, although regret about past actions should not be confused with self-stigma. Father involvement within the family is becoming an increased area of study due to its impact on wellbeing and child development, highlighting how it is not just involvement that is important but the quality of this involvement [31]. How fathers see themselves in this role, and how the parent identity interacts with other social identities is important to understand when exploring fathers within the family dynamic. Whilst a body of literature explores the negotiation of drug user identities by mothers [14,23,24], there is a paucity of literature on the identities of fathers [32,33], although this role was significant in both settings.

## 5. Conclusions

This article contributes to the literature on stigma in two important ways. Firstly, by comparing the stigma management strategies of those in active drug use and recovery, it explores individuals’ experiences of situated stigma at different stages of their drug use journeys. In doing so, it illustrates some of the similarities and differences in stigma management strategies, namely in relation to distancing, drug exceptionalism, and identity negotiation. Secondly, by taking an ethnographic approach to situated stigma, it has been possible to observe some of the nuances and subtleties in individuals’ experiences, and negotiations, of stigma. Significantly, this approach illustrates the pervasiveness of stigma throughout an individual’s drug use, as even those who are in recovery struggle to shake the residual stigma associated with once being an ‘addict’ or a ‘junkie’. This may partly be due to the stigma management strategies that compel individuals to merely displace stigma rather than resolve it.

The ethnographic observation of a homeless hostel and recovery service has allowed for the comparison of stigma within the interactions of people at different stages of their drug use journey. Our understanding of these differences is situational and therefore does not explain how and when the negotiation of stigma changes through an individual’s life course. Future research would benefit from additional ethnographic enquiry, to further comprehend the ways in which stigma arises, changes, and is managed *in situ*. It could also seek to address an identified gap in the literature, by exploring fatherhood within the context of past and/or present drug use.

## Data Availability

Due to the qualitative nature of this study, data sharing is not applicable due to ethical considerations, such as anonymity.

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
