# Peer review of "Situating Stigma: An Ethnographic Exploration of How Stigma Arises in Interactions at Different Stages of People’s Drug Use Journeys"

_ijerph, 2023, doi:10.3390/ijerph20196894_

Round 1

Reviewer 1 Report

I enjoyed reviewing this well written and interesting paper that offers a valuable contribution to the literature on stigma experienced by PWUD. I also congratulate the authors on completing an ethnographic study during the COVID pandemic.

Some language in the introduction could be clearer. It is uncertain who the article is written for, but more plain English may make it more accessible for non-researchers.

In the methods, the use of online zoom observation was interesting and I would have appreciated further details of how and when this occurred and whether participants were aware they were being observed.

Please provide details of the consent process for participants to participate. Were they given an information sheet and was participation voluntary? Was written consent obtained? Did any residents refuse to participate?

I would also have liked some further details regarding the 'creative timeline' and the 'photo elicitation' activities that are referred to. For example, a description of these activities, what were their aims, how were participants recruited, what data were collected, how and by whom, how were the data analysed?

Some information on the positionality of both doctoral students or the research team would have been useful to include and to understand any prior relationships between the researchers and the settings which may have influenced what was observed. 

Page 3 line 110 - it would be useful to know what the turnover of hostel residents is - average duration of stay - over the Jan 2020 - Nov 2021 period, what did the researchers do to make residents feel comfortable with the ethnographic process?

State that participant names are pseudonyms (assuming that they are).

It would be useful to have a summary of participant demographics - e.g. age or age range, gender, ethnicity, time in hostel if available. 

Include a footnote for lesser known drugs (to this reviewer anyway) e.g. spice. Spell out acronyms on first use e.g. IV. 

Page 4 line 181 - typo - once individuals move out of ...

Page 5 line 222 - reflecting on their own beliefs

Page 6 line 259 - add (pregabs) after pregablin since the short form is used in the quote that follows

P.6 line 277 - I found this observation really interesting
Therefore, a hierarchy of drugs is not used to ‘other’, but in respect of the perceived personal and social harms which they cause

Page 7 line 291 - add footnote for 'purple hearts'

Page 8 line 348 - some in text quotes are double quotations, others are single quote marks

Some commentary on the rigour of the study was missing, including strengths and limitations, how trustworthiness, credibility, and authenticity of the study were assured along with opportunities for transferability of findings.

Reviewer 2 Report

The aim of the article is to investigate how drug using people experience and negotiate stigma related to their drug use. Authors conducted ethnographic research in two different sites: a homeless hostel and a substance use center. The purpose of selecting these two sites is to compare the experience and negotiation of those who are in an active stage of their drug use (living in homeless hotel) and those who are in a recovery stage (being in a drug use service)

Drug use related stigmatization is well documented in the literature. What I liked in this article, and I think that it is partly its novelty, that authors reveal and document how drug users itself negotiate stigma in three different ways: „othering”, ’drug exceptionalism’ and ’negotiating identity’. They also tried to find differences between the clients of the two sites or between drug users in different stage of their drug use.

In detail:

The introduction chapter gives a proper overview of the literature. The cited references are relevant and published recently. This chapter well supports the aim of the authors, and this aim is clearly formulated.

The methodological chapter well reasons the usefulness of ethnographic method. But this chapter is a bit broad-brush. We can’t find out what was the aim of the different listed methods, how these methods contributed to the results. I miss some arguing why homeless hostel was the site of recruiting active drug users. I also miss some explanation how they could distinguish stigma related to drug use and stigma related to homelessness.

Results are correctly presented; quotations are interesting and well support authors’ assertions.

Discussion and conclusion refer to the presented results. The difference between clients of homeless hostel and drug treatment center in stigma management is not convincingly presented. Here they should also discuss the effects of homeless status on being stigmatized and better highlight the difference in the perception and negotiation of stigma between the two examined groups.

The importance of the paper is that it contributes to the better understanding of stigma management of drug users.

Reviewer 3 Report

This is an exceptionally well written and argued research paper demonstrating the importance of ethnographic observation for developing a better understanding of situated stigma in the lives of active substance users and substance users in recovery. The arguments developed are compelling and well situated in the theoretical and research literature on stigma. The presentation of the data draws on narratives in field notes from two studies to provide a nuanced understanding of how people who use drugs negotiate identities through othering, self-distancing, internalizing stigma, and creation of drug hierarchies of worthiness and risk. The collaboration is effective and significantly contributes to the literature on substance use and stigma through creation of a product that is greater that the sum of its two parts. 

A suggestion for improvement of the manuscript stems from the authors’ emphasis on better understanding the negotiation of identities by fathers, which is noted to be relatively neglected as compared to the role conflict experienced by substance using mothers. This is an accurate assessment of the literature worth building on with reference to whatever contributions may exist. A quick scan on Google Scholar yields the following examples that are worth reviewing for the insights they may yield:

Neault, N., Mullany, B., Powers, J., Coho-Mescal, V., Parker, S., Walkup, J., Barlow, A. and Cradling Our Future and Focus on Fathers Study Teams, 2012. Fatherhood roles and drug use among young American Indian men. The American journal of drug and alcohol abuse38(5), pp.395-402.

Taylor, M., 2012. Problem drug use and fatherhood (Doctoral dissertation, University of Glasgow).

It may be also worth considering the more extensive research literature on fathering for further reference to better situate these findings as an important area of study. See, for example:

Diniz, E., Brandão, T., Monteiro, L. and Veríssimo, M. (2021), Father Involvement During Early Childhood: A Systematic Review of the Literature. J Fam Theory Rev, 13: 77-99.
